# NaN Pooling & Convolution Accelerate U-Nets

## Abstract

Recent advancements in deep learning for neuroimaging have resulted in the development of increasingly complex models designed for a wide range of tasks. Despite significant improvements in hardware, enhancing inference and training times for these models remains crucial. Through a numerical analysis of convolutional neural networks (CNNs) inference, we found that a substantial amount of operations in these models are applied to pure numerical noise, with little to no impact on the final output. As a result, some CNNs consume up to two-thirds of their floating-point operations unnecessarily.

To address this inefficiency, we introduce NaN Pooling & Convolution—novel variations of PyTorch's max pooling and 2D convolution operations. These techniques identify numerically unstable voxels and replace them with NaNs, allowing models to bypass operations on irrelevant data. We evaluate NaN Pooling and Convolution on two models: the FastSurfer CNN, a widely used neuroimaging tool, and a CNN designed to classify the MNIST dataset. For FastSurfer, our approach significantly improves computational efficiency, skipping between 33.24% and 69.30% of convolutions in certain layers while preserving the model's original accuracy. On MNIST, our approach skips up to 28.38% of convolutions, again without major impact on the accuracy.

## 1 Introduction

Convolutional Neural Networks (CNNs), in particular U-Nets, are transforming neuroimaging by progressively replacing traditional image analysis software with models that deliver comparable performance in a fraction of the runtime. This advancement significantly enhances the field, enabling the routine processing of larger databases in reasonable timeframes. However, optimizing the inference and training times of these models remains a critical challenge, as improvements in this area could facilitate near-real-time analyses across various applications and support the training of larger models for tasks currently unattainable with existing approaches.

Through previous investigations of numerical stability in CNNs, summarized in Appendix A.5, we identified a numerical instability in the max pooling operation that leads to the propagation of pure numerical noise in approximately two thirds of the embedding values. Consequently, a substantial number of operations—particularly convolutions—end up only processing this noise and could be eliminated to improve efficiency.

The source of this instability is illustrated in Figure 1. When the forward calculation of the max pooling operation is applied to a relatively uniform window—where multiple values can achieve the maximum up to an epsilon—the position of the max index becomes undetermined. Multiple values in this window are now eligible for being the maximum value and they each have a different index. This has no immediate impact, but when the unpooling operation is called, the instability arises. Unpooling uses the indices saved from the max pooling operation to restore the maximum values to their original positions, filling the remaining voxels with zeros. The undetermination in the max pooling operation leads to several values in the unpooling operation being assigned either a zero or a non-zero value, resulting in a total loss of numerical precision.

Surprisingly, models affected by this numerical issue can still be trained and produce accurate results despite the widespread propagation of numerical noise. This suggests that the impacted values

do not contribute to the model's output, revealing a potential avenue for enhancing computational efficiency.

To capitalize on this observation, the execution framework must account for numerical precision, allowing operations on numerical noise to be bypassed. However, accurately measuring and representing numerical precision at the scale of CNN executions is impractical and would significantly slow down processing. Instead, we represent values with no numerical significance using IEEE NaN values, which are already supported by existing frameworks. We have modified the max pooling operation to generate NaNs in case of numerical instability, and we adjusted the convolution operation to handle tensors containing NaNs, bypassing computations when NaNs exceed a specified threshold.

We tested NaN Pooling on two CNNs: FastSurfer, a widely-used neuroimaging U-Net for whole-brain segmentation Henschel et al. (2020), and a CNN classifying the MNIST dataset LeCun & Cortes (1998). Our results demonstrate that for FastSurfer, NaN Pooling can bypass up to 69.30% convolutions in some layers, and up to 44% convolutions in a full model while maintaining the accuracy of model outputs. For MNIST, NaN Pooling skips up to 28.38% convolutions, again while achieving comparable accuracy.

## 2 METHODS

### 2.1 NUMERICAL INSTABILITY IN MAX POOLING

Max pooling Boureau et al. (2010) is a widely used downsampling technique that replaces a defined window of values with its maximum value. It can optionally return indices that indicate the original locations of these maximum values. During upsampling, max unpooling uses these indices to restore the maximum values to their original positions, filling the remaining voxels with zeros. This process ensures that the spatial structure of the input data is partially reconstructed based on the locations of the selected maximum values. The indices generated during max pooling are especially useful in U-Net architectures, where downsampling and upsampling processes are frequently coupled Zeiler et al. (2010); Çiçek et al. (2016); Lu et al. (2019); Plascencia et al. (2023); De Feo et al. (2021).

In our previous work Anonymized, we investigated the numerical uncertainty of CNNs during inference. We found that numerical instabilities arose during max unpooling operations due to fluctuations in the indices passed to this process. When values within a pooling window are close to each other, even slight noise—introduced, for example, by variations in the execution environment—can lead to index shifts while the maximum value remains unchanged. This instability is particularly evident when upsampling is applied to areas of an image's background, where uniform values prevail. Interestingly, we observed that the propagation of this numerical noise did not adversely impact the final outputs of the models we tested.

Unstable voxels contribute no meaningful information to the model. To address this inefficiency, we propose NaN Pooling and Convolution as a way to bypass operations on such irrelevant voxels. In floating-point arithmetic, NaNs (Not-a-Number) are special values defined by the IEEE 754 standard to represent undefined or unrepresentable results, such as 0 divided by 0 or the square root of a negative number. A NaN is represented by an exponent of all ones and a non-zero mantissa, and it is used to flag errors or exceptional conditions in calculations. Leveraging this concept, we use NaNs to mark numerically irrelevant voxels, effectively skipping over operations that would otherwise be wasted on data that provides no useful information. This approach enhances computational efficiency by allowing the model to focus on relevant data, without altering the final output or model performance.

### 2.2 NAN POOLING

As max pooling is the origin of the numerical uncertainty in U-Nets, we propose NaN Pooling to address the inefficiencies found. Below we define NaN Pooling and illustrate it in Figure 1.

First we define max pooling, where for each tensor window $\mathbf{W}$ in the input tensor $\mathbf{X}$, the max pooling operation $Y(\mathbf{W})$ is computed per batch.

$$Y(\mathbf{W}) = (m, i_m)$$

Figure 1: Comparison of Max Pooling vs NaN Pooling in the presence of numerical uncertainty. Green color represents numerically stable values, while red represents unstable values.

Where $m$ is the maximum value of $\mathbf{W}$ and $i_m$, the index of the maximum value of $\mathbf{W}$, i.e. $i_m = \text{argmax}(\mathbf{W})$.

For NaN pooling, we redefine the max pooling operation $Y$ to handle potential NaNs and tie-breaking for repeated maximum values as follows:

$$Y'(\mathbf{W}) = \begin{cases} (NaN, (0,0)) & \text{if } \text{Count}(\{\mathbf{W}_{:,:,j}\,,\, |\mathbf{W}_{:,:,j} - m| < \epsilon\}) > t_1 \\ (m, i_m) & \text{otherwise} \end{cases}$$

Where $t_1$ is a user-defined threshold that specifies the maximum number of near-equal values allowed for $\bar{m}$, and $\epsilon$ is a small tolerance set to $10^{-7}$ to handle floating-point precision issues. We set (0,0) to be the index in the presence of NaNs, because it is the simplest, most efficient and most stable value to implement when resetting indices. Should $\mathbf{W}$ contain NaN values, we simply ignore them when calculating $m$ and $i_m$.

## 2.3 NaN Convolution

NaN Convolution handles the presence of NaNs introduced through NaN Pooling, skipping over numerically irrelevant operations. Consider a padded 4D input tensor $\mathbf{X}$ of shape $(N, C_{in}, H_{in}, W_{in})$, a 4D kernel tensor $\mathbf{K}$ of shape $(C_{out}, C_{in}, H_k, W_k)$, and a NaN threshold $t_2 \in [0, 1]$ , where $N$ is the batch size, $C_{in}$ is the number of input channels, $C_{out}$ is the number of output channels, $H_{in}$ is the height of the input, $W_{in}$ is the width of the input, $H_k$ is the height of the kernel, and $W_k$ is the width of the kernel.

For each window $\mathbf{W}$ in the input tensor, where $\mathbf{W}$ is of shape $(C_{in}, H_{in}, W_{in})$ and its elements are in $\mathbb{R} \cup \{NaN\}$, we define the output of the NaN convolution of window W by kernel $\mathbf{K}$ as performed per batch:

$$Y_{c,h,w} = \begin{cases} NaN & \text{if } r_{c,h,w} \geq t_2 \\ \sum_{c=0}^{C_{in}-1} \sum_{h=0}^{H_k-1} \sum_{w=0}^{W_k-1} \bar{W}_{c,h,w}\, K_{c,h,w} & \text{if } r_{c,h,w} < t_2 \end{cases}$$

Where $r_{c,h,w}$ is the total number of NaNs across the input channels, height and width dimensions:

$$r_{c,h,w} = \frac{\text{Count}(\{w \in \mathbf{W}_{n,i,j}, w = NaN\})}{C_{in} H_{in} W_{in}}$$

We define $\bar{\mathbf{W}}$ as the modified window where NaNs are replaced with one of two approaches.

Approach A replaces NaNs with $\mu_{n,i,j}$, defined as the mean of the non-NaN values within $\mathbf{W}$:

$$\bar{\mathbf{W}} = \begin{cases} \mu_{n,i,j} & \text{if } \mathbf{W}_{n,i,j} = NaN \\ \mathbf{W}_{n,i,j} & \text{otherwise} \end{cases}$$

Approach B replaces NaNs with a random value generated from a Gaussian distribution centered around $\max_{n,i,j}$, defined as the maximum of the non-NaN values within $\mathbf{W}$, and a standard deviation

$\sigma$ of $10^{-3}$ .

$$\bar{\mathbf{W}} = \begin{cases} x \sim \mathcal{N}\left(\max_{n,i,j}(\mathbf{W}), \sigma\right) & \text{if } \mathbf{W}_{n,i,j} = NaN \\ \mathbf{W}_{n,i,j} & \text{otherwise} \end{cases}$$

Approach A can smooth the output of the NaN Convolution, which is occasionally advantageous. However, when smoothing is undesirable, Approach B introduces variability into the output. This variability is particularly useful in models with subsequent iterations of NaN Pooling, as it prevents overly aggressive NaN introduction that could result from repeatedly exploiting the smoothed output of Approach A.

$\bar{\mathbf{W}}$ is introduced to ensure that regions where the number of NaNs remains below the threshold $t_2$ are unaffected, since standard deep learning operations cannot inherently manage NaN values. It replaces the previous versions of the window and serves as the basis for the convolution operation.

## 2.4 NaN Convolution Implementation

Implementing NaN Convolution requires modifying PyTorch's internal handling of 2D convolutions. Instead of processing convolutions on a window-by-window basis, PyTorch executes an entire convolution layer as a single matrix multiplication. This formulation requires us to adapt the NaN Convolution definition to align with PyTorch's more efficient computational strategy.

PyTorch's implementation of 2D convolutions lies in the im2col technique, illustrated in the Appendix Figure 7. This operation reshapes the input tensor by extracting overlapping subregions (corresponding to the convolutional kernel's dimensions) and arranges them into columns, which can then be multiplied with the kernel weights in a single matrix multiplication step. To reduce the occurrence of random memory seeks, the im2col process is typically column-major, meaning data is stored and processed primarily by columns. This technique dramatically increases computational efficiency compared to the naive approach Chetlur et al. (2014).

For an input of size $(N, C, H_{in}, W_{in})$ and a kernel of size $(C, H_k, W_k)$, im2col extracts subregions of size $(C, H_k, W_k)$ and arranges them into columns, repeating the process for each of the $N$ batches. In our NaN Convolution implementation, we reduce the number of columns by removing subregions that exceed a predefined NaN threshold. By eliminating these irrelevant columns, we reduce the memory footprint and the computational load, thus improving overall performance. This optimization aligns with our theoretical expectations, offering a practical approach to reducing the computational inefficiencies while preserving model accuracy.

Given a padded 4D input tensor $\mathbf{X}$ of shape $(N, C_{in}, H_{in}, W_{in})$, a 4D kernel tensor $\mathbf{K}$ of shape $(C_{out}, C_{in}, H_k, W_k)$, and a NaN threshold $t_2 \in [0, 1]$ , we define $N$ as the batch size, $C_{in}$ as the number of input channels, $C_{out}$ as the number of output channels, $H_{in}$ as the height of the input, $W_{in}$ as the width of the input, $H_k$ is the height of the kernel and $W_k$ as the width of the kernel.

We unfold the input tensor $\mathbf{X}$ using PyTorch's unfold operation, in order to obtain matrix $M \in \mathbb{R}^{(N \times H_{in} \times W_{in} , C_{in} \times H_k \times W_k)}$. We then rename $M$ to $\mathring{M}$, to indicate the possible presence of NaNs and to calculate the NaN ratio $r_j$:

$$r_j = \frac{\text{Count}(\{m \in \mathring{M}_{:,j}, m = NaN\})}{C_{in} H_{in} W_{in}}$$

We then remove columns from $\mathring{M}$ that surpass the user set NaN threshold, $t_2$, and name this truncated matrix $\mathring{M}_{trunc}$:

$$\mathring{M}_{trunc} = \{\mathring{M}_{:,j} \mid r_j \leq t_2, j = 0, 1, ..., C_{in} H_k W_k\}$$

$\bar{M}_{trunc}$ is built from $\mathring{M}_{trunc}$ as it has all remaining NaN values that are under $t_2$ replaced with the mean value of their respective column. This is done to preserve standard convolutional operations which cannot inherently manage NaN values.

$$\bar{M}_{trunc}[i,j] = \begin{cases} \mathring{M}_{trunc}[i,j] & \text{if } \mathring{M}_{trunc}[i,j] \text{ is not } NaN \\ \text{mean}(\mathring{M}_{trunc}[:,j]) & \text{if } \mathring{M}_{trunc}[i,j] \text{ is } NaN, \end{cases}$$

We then perform the matrix multiplication:

$$\bar{Y}_{trunc} = \bar{M}_{trunc}\, K_{unfold}$$

Where $K_{unfold} \in \mathbb{R}^{(C_{out}, C_{in} \times H_k \times W_k)}$ and $\bar{Y}_{trunc}$ is the output of the truncated matrix multiplication.

Finally, we add the removed columns back to their original locations in $\bar{Y}_{trunc}$, populated solely with NaNs in order to obtain $\bar{\bar{Y}}$, of shape $(N, C_{out}, H_k, W_k)$.

In our implementation of NaN Convolution, we allow users to set the NaN threshold $t_2$, providing control over how aggressively NaNs are managed during operations. A higher threshold reduces the occurrence of NaNs, which can lead to reduced computational efficiency but thoroughly maintains model performance. Adjusting the threshold often involves a trade-off between efficiency and accuracy, as allowing more NaNs can impact key operations and potentially degrade performance.

## 3 EXPERIMENTS

We evaluated NaN Pooling and Convolution on two different CNNs. The first use case is a popular neuroimaging U-Net, FastSurfer, using a representative set of images from a dataset provided by the Consortium for Reliability and Reproducibility (CoRR) Zuo et al. (2014). Evaluation metrics for FastSurfer included the ratio of skipped convolutions, and the loss functions utilized to train the original model. The second CNN is built to perform digit classification on the MNIST dataset Le-Cun & Cortes (1998). It is a widely known task that showcases NaN Pooling and Convolutions' applicability beyond neuroimaging specific models.

### 3.1 FASTSURFER

FastSurfer is a CNN model that performs whole-brain segmentation, cortical surface reconstruction, fast spherical mapping, and cortical thickness analysis. The FastSurfer CNN is inspired from the QuickNAT model Roy et al. (2019), which is composed of three 2D fully convolutional neural networks—each associated with a different 2D slice orientation—that each have the same encoder/decoder U-net architecture with skip connections, unpooling layers and dense connections as QuickNAT. A diagram of the model's architecture is available in the Appendix Figure 8. The FastSurfer segmentations were shown to surpass state-of-the-art methods, as well as being generalizable to unseen datasets and having better test-retest reliability. We used the pre-trained model from FastSurfer available on GitHub fas. We focus exclusively on the task of whole-brain segmentation, defined as the labeling of different anatomical brain regions, which is performed solely by the CNN.

### 3.2 MNIST

To evaluate the performance and behavior of NaN Pooling and NaN Convolution, we conducted experiments on a small CNN trained on the MNIST dataset. The model architecture included alternating convolutional and ReLU activation layers, each followed by a pooling layer, repeated three times, and concluded with a final convolutional layer, a pooling layer, and a log-softmax output layer. While this architecture is not a U-Net — considered the ideal setting for applying NaN Pooling and Convolutions — we show that this approach is also effective for other CNN architectures utilizing convolution and max pooling operations. The task, a classification problem to identify digits from the input images, is well-established and widely considered solved. This experiment served as a baseline to demonstrate the feasibility and potential benefits of NaN Pooling in a controlled, well-understood context.

### 3.3 DATASET & PROCESSING

For FastSurfer, we used the Consortium for Reliability and Reproducibility (CoRR) dataset, a multi-centric, open resource aimed to evaluate test-retest reliability and reproducibility. We randomly selected 5 T1-weighted MRIs from 5 different subjects, one from each CoRR acquisition site, and accessed them through Datalad Halchenko et al. (2021). The selected images included a range of image dimensions, voxel resolutions and data types (Ap-

pendix A.2). We processed all subjects' images using FreeSurfer's recon-all command with the following steps: `--motioncor --talairach --nuintensitycor --normalization --skullstrip --gcareg --canorm --careg`. These steps ensured that the images were motion-corrected, skull-stripped, intensity-normalized, and registered both linearly and non-linearly, preparing them as input for FastSurfer segmentation.

For the MNIST use case, the CNN used in this experiment was custom-built while the dataset was downloaded from PyTorch's torchvision library.

When applying NaN Pooling and Convolutions to the models, we used Approach A within NaN Convolution for NaN substitution for the FastSurfer CNN and Approach B for the MNIST CNN in order to achieve optimal performance.

We processed the data for FastSurfer on the Narval cluster from École de Technologie Supérieure (ETS, Montréal), managed by Calcul Québec and The Digital Alliance of Canada which include AMD Rome 7502, AMD Rome 7532, and AMD Milan 7413 CPUs with 48 to 64 physical cores, 249 GB to 4000 GB of RAM and Linux kernel 3.10. We executed the MNIST CNN on the slashbin cluster with $8 \times$ compute nodes each with an Intel Xeon Gold 6130 CPU, 250 GB of RAM, and Linux kernel 4.18.0-240.1.1.el8_lustre.x86 64. We used FreeSurfer v7.3.1, FastSurfer v2.1.1, PyTorch v2.4.0, and Singularity/Apptainer v1.2. The scripts and documentation for this experiment will be published on GitHub.

## 4 RESULTS

### 4.1 NaN POOLING AND CONVOLUTION SAVES 39% OF CONVOLUTIONS ON AVERAGE

To quantify the acceleration introduced by NaN Pooling and Convolution, we measured the number of convolutional operations replaced with NaNs in the FastSurfer model. We tested the techniques with several thresholds ranging from 1 to 0.5 depending on the use case. The threshold represents the ratio of NaNs required to skip an operation, with threshold 1 being the most stringent (skipping operations only when the input consists entirely of NaNs) and threshold 0.5 being more lenient (skipping operations when 50% or more of the values are NaNs).

Our results revealed that the numerically unstable voxels impacted by NaN Pooling and Convolution are typically found in the background of the input data. This often numerically irrelevant background can comprise up to two-thirds of the total input space, rendering it ideal for optimization. To quantify the computational impact, we calculated the ratio of skipped operations relative to the total number of convolutional operations. This ratio was tracked both across the architectural layers of the models and across brain slices in the neuroimaging data, providing a comprehensive view of the effect of NaN Pooling and Convolution.

Figure 2 shows the impact of NaN Pooling and Convolution on FastSurfer during inference. As illustrated in Figure 2a, skipped operations only occur between the Encode 1 and Decode 2 blocks. This is because no NaN Pooling is applied prior to Encode 1, and no NaN Pooling occurs after the Bottleneck layer, leading to a decline in skipped operations after Decode 2 as NaNs become sparse. As a result, the majority of skipped operations are concentrated in the middle sections of the model with the exception of the Bottleneck block. The drop here is due to the U-Net architecture's reduced spatial dimensions at this stage, which prioritises the preservation of relevant information while minimising the presence of numerically irrelevant voxels (represented by NaNs).

In Figure 2b, we observe higher skipped operation ratios at the extremes of the brain slice distribution, which implies a higher amount of numerically irrelevant voxels. Upon examination, these slices contain a larger proportion of background than brain matter voxels. This is consistent with preliminary results which suggested background voxels are largely numerically irrelevant.

For FastSurfer, NaN Pooling and Convolution reduced the number of operations by 33.24% and 44.19% at thresholds of 1 and 0.5, respectively. When focusing solely on model layers affected by NaNs, the skipped operations increased significantly to 50.59% and 69.30% for these thresholds.

In brief, the most substantial computational gains were observed in the encoder section of the U-Net architecture, largely due to the frequent use of NaN Pooling. This technique primarily targeted

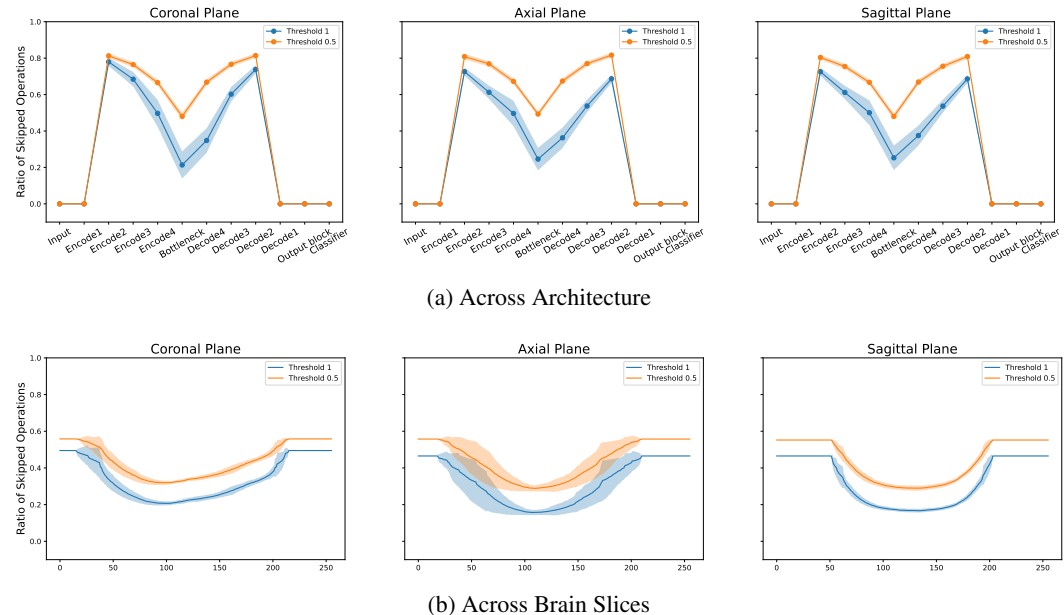

Figure 2: Ratio of Skipped Convolutions Across FastSurfer for Architecture and Brain Slices. For Threshold 1 (blue) and 0.5 (orange), 33.24 % and 44.19 % of total convolutional operations were skipped, respectively.

numerically irrelevant voxels, which were concentrated in background regions or areas devoid of brain matter.

## 4.2 NaN Pooling and Convolution Preserves Model Accuracy

Besides optimizing the computational efficiency of the model, we want to maintain its current performance. We evaluate the models' performance with NaN Pooling and Convolution against the model run with their default operations using metrics commonly used to evaluate brain segmentation as well as metrics used to evaluate the original performance of the models.

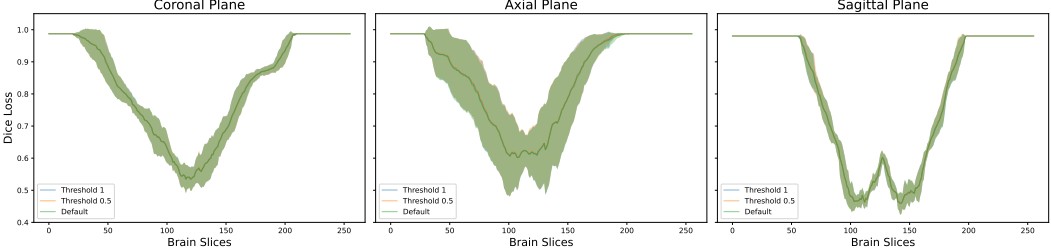

Figure 3: Comparison of Dice loss for default and NaN FastSurfer (thresholds 1 and 0.5) across brain slices.

We begin our analysis by examining the distribution of Dice loss across brain slices, as shown in Figure 3. Understanding Dice loss requires a grasp of the Dice coefficient score, which measures the overlap similarity between two brain segmentations. Higher Dice coefficients indicate more similar segmentations, while Dice loss is calculated as one minus the Dice coefficient. Consequently, higher Dice loss values signify less similarity between predicted and ground truth segmentations.

The average and standard deviation of Dice loss across subjects for different thresholds closely resemble those of default FastSurfer. As expected, we observe a decrease in Dice loss near the centre of the brain, where slices contain the highest proportion of brain matter, resulting in greater

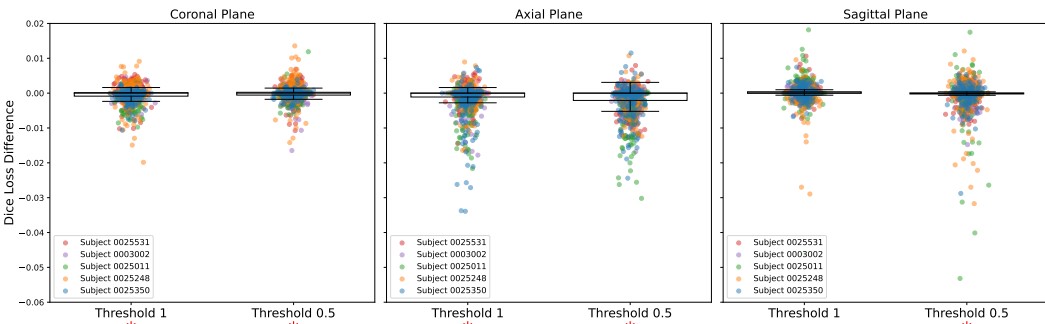

Figure 4: Comparison of Dice loss differences between NaN-FastSurfer and default FastSurfer across brain planes and thresholds. Significant differences between NaN and default FastSurfer as calculated with a paired T-test are indicated by *.

similarity. In contrast, slices with more background show lower similarity and higher Dice loss. However, a sharp increase in Dice loss is observed near the centre of the distribution for the sagittal plane. This increase is likely attributed to the presence of the longitudinal fissure (or sagittal fissure) that separates the two brain hemispheres. The two hemispheres are primarily connected by the corpus callosum, and the empty space in this fissure leads to increased Dice loss.

Expanding on this analysis, Figure 4 shows the Dice loss differences between NaN-FastSurfer and default FastSurfer across subjects. Negative values indicate worse performance for NaN-FastSurfer, while positive values indicate improvement. Overall, NaN-FastSurfer performs similarly to the default model, with differences tightly clustered around zero across the three brain planes.

At threshold 1, significant differences are seen in the coronal and axial planes, while at threshold 0.5, they appear across all planes. The largest difference (-0.05) is found in the sagittal plane at threshold 0.5, representing a 5.73% variation from its default Dice score. Excluding outliers, average Dice differences are 0.02% and 0.04% for thresholds 1 and 0.5, respectively, rising slightly to 0.06% and 0.09% when outliers are included.

Nonetheless, we also observed a substantial number of negative differences between NaN and default FastSurfer in the axial plane, as depicted in Figure 3. This variability is further illustrated in Figure 4, where the axial plane exhibits a high standard deviation, particularly in comparison to the coronal and sagittal planes. Upon further investigation into the source of these differences, Figure 5 highlights that the cerebellum is a significant contributor to the variability between the two methods. Although the cerebellum is not visible in this axial slice, it becomes particularly evident in lower axial slices and demonstrates instability across subjects and methods. Research has indicated that the cerebellum is notoriously challenging to segment due to its complex anatomy, proximity to other brain regions, high shape variability across subjects, and often low contrast in neuroimaging data, which complicates detail identification. Furthermore, FastSurfer was trained on FreeSurfer segmentations, which has been noted for its poor segmentation performance in regions of low contrast and intricate anatomy Morell-Ortega et al. (2024); Carass et al. (2018); Romero et al. (2017). Additional visualisation of the expected cerebellum quality is provided in the Appendix Figure 9. While we observe a visual decline in quality with NaN-FastSurfer, the default version is not considered a valid segmentation either. This supports the conclusion that the limitations of FreeSurfer extend to FastSurfer, rendering its cerebellum segmentation unreliable. Aside from the cerebellum, the segmentation quality of the rest of the model appears stable and consistent between NaN and default FastSurfer.

Interestingly, we found that the performance of NaN-FastSurfer is quite similar between thresholds 1 and 0.5. Although we observed slightly more differences from the default FastSurfer at threshold 0.5, the overall variations between the two thresholds were minimal. Therefore, while further testing is needed to generalise our findings to other use cases, we can conclude that threshold 0.5 is the preferred option, as it achieves comparable performance with enhanced computational efficiency.

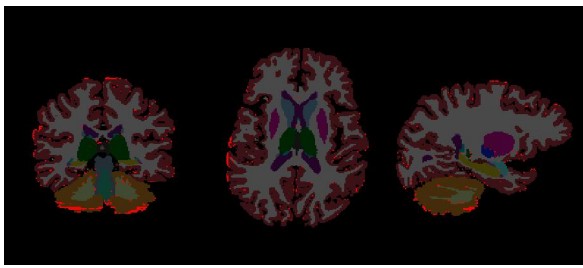

(a) Worst Performing Subject for Threshold 1; sub-0025011

(b) Worst Performing Subject for Threshold 0.5; sub-0025011

Figure 5: Comparison of segmentation outputs between NaN-FastSurfer and default FastSurfer across different thresholds, displayed in coronal (left), axial (center), and sagittal (right) planes. The different brain regions are colored according to the Fastsurfer colormap except for the bright red voxels scattered throughout the brain which denote differences in segmentation outputs.

## 4.3 THRESHOLD SENSITIVITY ANALYSIS WITH MNIST

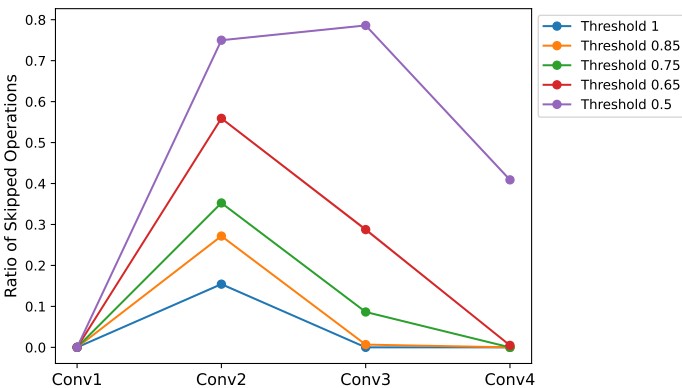

Figure 6: Ratio of Skipped Convolutions Across MNIST CNN Architecture for Test Dataset.

Table 1: MNIST CNN performance metrics with Kfold validation (K=5) across different thresholds and the default model.

|  | Threshold 1 | Threshold 0.85 | Threshold 0.75 | Threshold 0.65 | Threshold 0.5 | Default |
|---|---|---|---|---|---|---|
| Precision | $0.988 \pm 0.001$ | $0.989 \pm 0.002$ | $0.982 \pm 0.003$ | $0.932 \pm 0.004$ | $0.304 \pm 0.028$ | $0.991 \pm 0.001$ |
| Recall | $0.989 \pm 0.001$ | $0.989 \pm 0.002$ | $0.981 \pm 0.003$ | $0.927 \pm 0.005$ | $0.198 \pm 0.006$ | $0.991 \pm 0.001$ |
| F1 | $0.989 \pm 0.001$ | $0.989 \pm 0.002$ | $0.981 \pm 0.003$ | $0.927 \pm 0.005$ | $0.194 \pm 0.007$ | $0.991 \pm 0.001$ |
| Accuracy | $0.989 \pm 0.001$ | $0.989 \pm 0.002$ | $0.981 \pm 0.003$ | $0.927 \pm 0.005$ | $0.198 \pm 0.006$ | $0.991 \pm 0.001$ |

Figure 6 illustrates the ratio of skipped operations during inference for the MNIST CNN. Skipped operations are observed only after the Conv1 layer, due to the introduction of pooling operations. Notably, the skipped operation ratio drops back to zero after the Conv4 layer. This is likely to happen because the Conv4 layer, being the most downsampled, primarily contains crucial information necessary for classification, with irrelevant information effectively filtered out by this stage.

For this model, we experimented with a wider range of thresholds, as the impact of skipping operations on model performance was significantly more pronounced compared to what was observed with FastSurfer. As shown in Figure 6, lower thresholds result in greater computational gains. Specifically, after incorporating pooling operations, the percentage of skipped operations increases as thresholds decrease: 5.14% for a threshold of 1, 9.28% for 0.85, 14.63% for 0.75, 28.38% for 0.65, and 64.83% for 0.5. However, these gains come at the cost of a trade-off in model performance, highlighting the importance of selecting an appropriate threshold.

Analyzing the results in Table 1, we observe that most thresholds for the NaN-enhanced MNIST CNN preserve comparable performance. However, the choice of threshold emerges as a critical factor for this model. Specifically, the most stringent threshold tested, 0.5, significantly degrades performance, while all other thresholds maintain metrics exceeding 90%.

Figure 6 and Table 1 highlight the importance of carefully selecting a threshold to balance performance and computational efficiency. This trade-off should be tailored to the specific requirements of the intended use case.

## 5 CONCLUSION

This paper introduced NaN Pooling and Convolution, new variations of PyTorch's max pooling and 2D convolution operations, which are designed to enhance the efficiency of U-Net models. The techniques identify and leverage numerically unstable voxels, which don't contribute to the model's output, by replacing them with NaNs. This allows the model to bypass operations on irrelevant data, thereby saving computation time. These potential benefits aren't limited to neuroimaging and could be advantageous in other fields facing similar challenges related to computational demands and data efficiency.

We evaluated the effectiveness of these methods on two use cases: the widely used neuroimaging U-Net, FastSurfer, and the MNIST benchmark CNN. For FastSurfer, our NaN Pooling and Convolution techniques achieved a 39% reduction in total convolutional operations on average, significantly improving computational efficiency without compromising model accuracy. The NaN-FastSurfer demonstrated performance comparable to its default implementation. Similarly, in the MNIST benchmark CNN, we achieved up to a 28.38% reduction in convolutional operations while maintaining comparable accuracy. However, the MNIST model showed greater sensitivity to threshold variations compared to FastSurfer, highlighting the importance of carefully balancing efficiency and performance when determining the optimal threshold.

Despite successfully skipping operations, we have not yet observed a direct impact on runtime speed-up, likely due to PyTorch's inherent computational optimizations. As a result, our analysis focuses on the number of skipped convolutional operations as a proxy for computational efficiency. Therefore, future work will focus on extending the application of NaN Pooling and Convolution to the training of models and delivering the expected runtime speed-ups from skipped operations in practice. Further investigation is required to evaluate potential hardware limitations, implementation-specific factors or investigating alternative implementations that leverage sparse matrix representations or specific hardware architectures.

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

# A APPENDIX

## A.1 IM2COL OPERATION

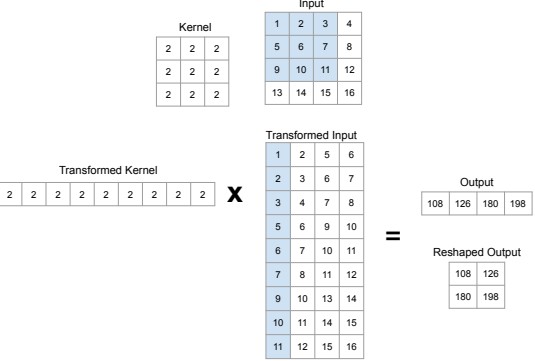

Figure 7: Illustration of PyTorch's im2col operation for an 4x4 input and 3x3 kernel to produce 2x2 output. The blue highlighted window indicates how the operation reshapes the each convolution window into a column.

## A.2 CORR DATASET QUALITY CONTROL

Table 2: Subjects sampled in the CoRR dataset.

| Subject | Image Dimension | Voxel Resolution | Data Type | Processing Status |
|---------|-----------------|------------------|-----------|-------------------|
| sub-0025248 | (208, 256, 176) | (1.00, 1.00, 1.00) | float32 | Success |
| sub-0025531 | (160, 240, 256) | (1.20, 0.94, 0.94) | float32 | Success |
| sub-0025011 | (128, 256, 256) | (1.33, 1.00, 1.00) | float32 | Success |
| sub-0003002 | (176, 256, 256) | (1.00, 1.00, 1.00) | int16 | Success |
| sub-0025350 | (256, 256, 220) | (0.94, 0.94, 1.00) | float32 | Success |

## A.3 U-NET ARCHITECTURE

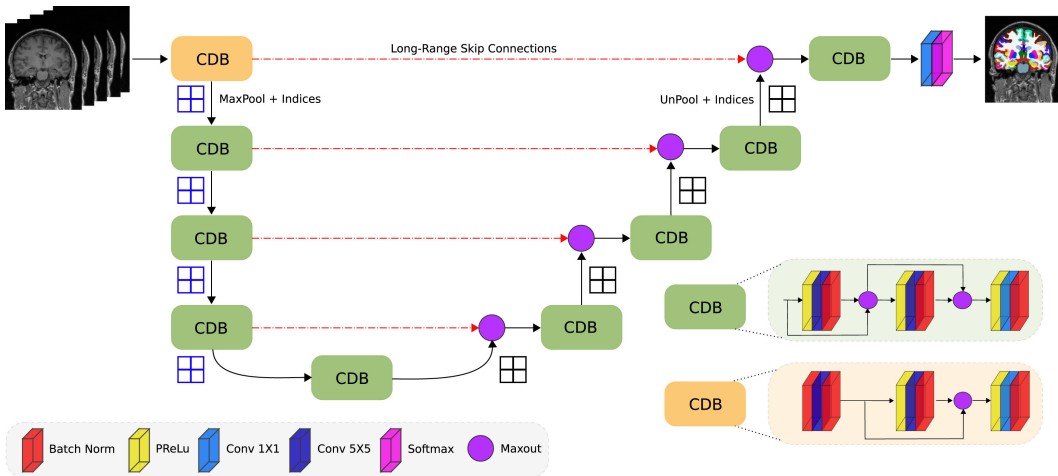

Figure 8: Illustration of FastSurfer's architecture. The CNN consists of four competitive dense blocks (CDB) in the encoder and decoder part, separated by a bottleneck layer. Figure reproduced from Henschel et al. (2020).

## A.4 CEREBELLUM SEGMENTATION

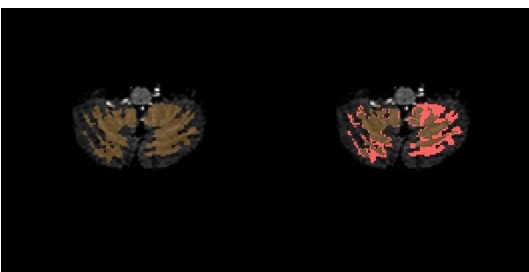

Figure 9: Comparison of FastSurfer's cerebellum segmentation with and without NaN Pooling and Convolution. On the left is the default FastSurfer segmentation, while on the right, the overlay shows the differences between NaN-FastSurfer (threshold 1) and the default version. Both segmentations are superimposed on the anatomical MRI scan of the cerebellum for reference.

## A.5 FASTSURFER UNSTABLE MODEL EMBEDDINGS

In our previous work Anonymized, we examined the numerical uncertainty of the FastSurfer CNN during inference, focusing on the stability of the final classification results as well as the embeddings.

In order to do so, we used Monte Carlo Arithmetic (MCA) Parker (1997), a stochastic arithmetic technique that introduces randomness to assess numerical uncertainty. MCA is implemented through the Verrou tool Févotte & Lathuiliere (2016), a tool that leverages Valgrind Nethercote & Seward (2007) to dynamically instruments binary executables with MCA. Using Verrou, we instrumented FastSurfer inference to generate 10 iterations of the model's embeddings, each subjected to random perturbations introduced during execution. These perturbations allowed us to simulate and analyze numerical uncertainty inherent in the model. To quantify this uncertainty, we computed the number of significant digits across the 10 iterations. Significant digits Sohier et al. (2021) measure numerical uncertainty by determining the number of digits shared in common across multiple MCA samples for a given floating-point value. This analysis was performed for every layer of the FastSurfer model, with the results visualized as heatmaps in Figure10. This revealed that a large fraction of the model embeddings were purely numerical noise (zero significant digits), represented by red-colored regions in the figure.

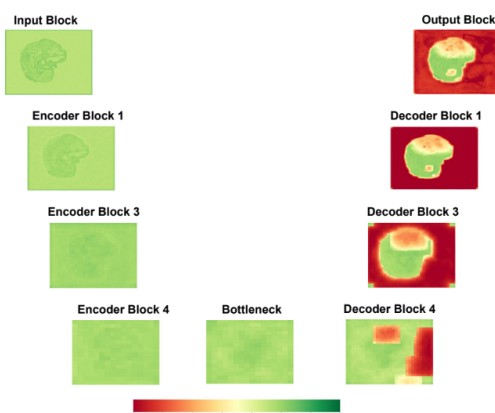

Figure 10: Significant Digit Maps for FastSurfer Model Embeddings in Selected Model Layers for Numerically Unstable Data.

The instability first appeared during the max unpooling operations in the FastSurfer decoder, which we later determined resulted from the indices provided to the max unpooling operation. This instability becomes especially pronounced when upsampling is applied to regions of the image background, where uniform values dominate.

Interestingly, the segmentations resulting from the model were still accurate in spite of the presence of substantial numerical noise in the embeddings, which suggested that computations performed on these values were not contributing to the final result. This observation motivated the design of NaN pooling and convolutions presented in this paper.

