# OpenReview forum: "NaN Pooling and Convolution Accelerate U-Nets"
_ICLR.cc/2025/Conference — Submitted to ICLR 2025_

### Official Review · Reviewer_mvfd · 2024-11-02

**Soundness:** 2
**Presentation:** 3
**Contribution:** 2
**Rating:** 3
**Confidence:** 4

**Summary:**

This paper introduces two operations: NaN pooling and NaN convolution to accelerate CNN inference speed. The authors demonstrate that, depending on the layers, convolution calculations can be saved from 33% to 69% while maintaining model accuracy.

**Strengths:**

1. the paper writing is mostly clear and easy to understand what the authors want to convey.

**Weaknesses:**

1. The authors should pay attention to equations, which should be part of the sentences/paragraphs; hence, punctuation and capital letters of the first word of the line after the equation, e.g., L128, L183-185, should be taken care of. Please also number the equations.
2. L043-044: "...where multiple values can achieve the maximum up to an epsilon—the position of the max index becomes undetermined." - please revise, this is not straightforward to understand until reading into the methodology. Also does it matter for the exact max index to be undermined? Max pooling handles translation invariance up to a small degree because of the pooling property.
3. L046-047: "This ambiguity leads to several values in the window produced by the max pooling operation being assigned either a zero or a non-zero value, resulting in a total loss of numerical precision." - the forward calculation of max pooling pool the values in a window to generate one value output per window, are you referring to the gradient of max pooling operation?
3. L046-047: "..this numerical ”bug” still yield meaningful results..." - even if the bug is in quotation marks, I don't agree it should be called like that, max pooling is intended to work in this way.
4.L104-105: in my opinion, I don't exactly understand the intuition if several values in a window are close to the maximum value of the window, then the output of the max pooling needs to be NaN. If as the author said, the numerical stability is an issue, shouldn't such a dramatic change in max pooling output cause a convergence issue? I can imagine such a pooling operation would give NaNs to image regions with very small intensity changes but give real value response to edges and local patterns, so essentially the operation is kind of letting the model focus on patterns instead of large color blobs.
5: L116: N is the batch size?
6: L120: why the window W contains the batch dimension N?
7. L133-136: what does the column refer to here? channel? If it is the same column definition in Sec 2.4, define it before use.
8: L134: \bar{W} redefines the window, does this \bar{W} replace the original window before the convolution? or this is the window on the next conv layer?
9: L134: \mu_{n,i,j} meant to be a batch-wise normalization? what is the intuition behind that?
10: L149-155: this could use a figure to illustrate the im2col, especially when you used columns a number of times.
11. the experiments are only on the CORR dataset--a 3D MRI brain segmentation dataset. The method seems to be general enough to test on computer vision datasets, and given this is ICLR, small datasets such as MNIST/CIFAR10/CUB etc would be OK to use as benchmarks. The choice of U-Net and segmentation tasks is also limited. Finally, since it is 3D data, it makes sense to consider implementing and testing 3D convolution kernels.
12. if the objective is to speed up, the authors should time the training/inference time not only the theoretical number of operations saved. I'd imagine the count operation introduces NaN and the mean operation that processes away the NaN would also take time to compute.

**Questions:**

I suppose the paper is clear about what the authors have done, the motivation starts from bypassing operations on the irrelevant data to improve computation efficiency. However, the experiments do not effectively test the objective.

---

### Official Review · Reviewer_3VPG · 2024-11-03

**Soundness:** 2
**Presentation:** 3
**Contribution:** 1
**Rating:** 3
**Confidence:** 5

**Summary:**

The paper identifies `max pooling` operation as a cause of instability when the max value in a window is not uniquely undeniable (there are multiple numbers in the window within an epsilon of the max value). Authors hypothesize that the instability is because the maximum index cannot be uniquely determined in this case. Authors propose "NaN Pooling" where the max pool operation is modified in output NaN values of the number of epsilon close max values in a window is greater than a user defined threshold. They pair it with NaN convolutions which is modified convolution operation that skips working on windows with more than a threshold of NaN values. Authors evaluate this methodology on FaserSurfer CNN and show that they skip about 37% of convolutions on an average without any loss in accuracy. Authors note that they do not see any loss in accuracy.

**Strengths:**

The paper is well written and easy to understand. The paper maybe more significant for a domain specific conference instead of a general ML conference like ML - please see sections below for more thoughts.

**Weaknesses:**

+ **Motivation for looking a the instability issue**

Authors mention that even though there is "numerical instability" it doesn't affect the accuracy or the final output quality. Then why did the authors notice this? The motivation for looking at this issue is unclear to me. Can authors motivate this more?

+ **Too domain specific/ Not a widespread problem?**

Authors solely focus on FastSurferCNN and just one dataset. Do authors believe that this is a more widespread problem? Can authors evaluate standard benchmark datasets like ImageNet and see if they observe the same phenomena?

The current motivation and evaluation setup seems to niche for an ICLR audience. If authors believe that this instability is general enough and their solution is general enough, I highly recommend that authors show this on multiple datasets and architectures.

+ **Results in the current form may not be significant**


Authors show an improvement in the number of convolution operations skipped. However, given the additional operations that is required to skip convolutions, I am unclear if this would translate to improvement in runtime even with optimizations as authors mention in the paper. Without an improvement in runtime, and the original instability not hurting accuracy, I am unclear on what is the significance of this result.

+ **Comparison with quantization/pruning methods**

Pruning methods can also remove filters reducing the number of convolutions. Can authors compare against off the shelf efficient methods to see how NaN Pooling + NaN Convolution compare?

**Questions:**

+ **Why use maxpooling**?

Line 042. The source of this instability is clearly identified. When the max pooling operation is applied to a relatively uniform window—where multiple values can achieve the maximum up to an epsilon - the position of max index becomes undetermined.

Why use max pooling in the first place? Can this not be replaced by, average pooling, for example?

+ **Why instability**?

In Line 0042-0043, author claim that the max value index cannot be uniquely determined and this causes instability. If the max values are epsilon close, why does it matter which index is picked as the max? As long as we pick any reasonable index, there should be no problem. Can authors explain on why this would create instability?

---

### Official Review · Reviewer_evGe · 2024-11-04

**Soundness:** 3
**Presentation:** 2
**Contribution:** 2
**Rating:** 3
**Confidence:** 5

**Summary:**

This paper introduces NaN Pooling and NaN Convolution, new methods designed to improve the efficiency of U-Net models by skipping operations on irrelevant data, identified as numerically unstable voxels, and replacing them with NaNs. Tested on FastSurfer, a widely-used neuroimaging U-Net model, these methods achieved a 39% reduction in convolutional operations without compromising accuracy. Although no direct runtime improvement was observed due to PyTorch’s optimizations, the reduction in operations demonstrates the potential for computational efficiency across various data-intensive applications.

**Strengths:**

- By identifying and skipping operations on irrelevant data, methods significantly reduce the number of convolutional operations.
- While tested on a neuroimaging U-Net, the methods seem to be broadly applicable.
- Despite the reduction in computations, the methods are claimed to maintain comparable model performance to the original.
- The methods introduce possibilities for further speed-up if combined with hardware-specific optimizations, such as sparse matrix operations or tailored architectures.

**Weaknesses:**

- The comparison of the results with various state-of-the-art and previous works is unclear.
- The proposed method does not seem to apply to any of the new architectures, such as transformers, which require high parallelization.
- The theoretical aspects of numerically unstable voxels and skipped convolutions are not discussed at all.
- There is only one limited experiment on a single dataset. It is not clear how the models work for other tasks such as classification and regression.
- It is not clear why there is no quantitative measurement of the accuracies on the actual 3D images. All evaluations appear to be done on each 2D projection of the three coronal, axial, and sagittal planes.

**Questions:**

- What are the colors in Fig. 4 representing?
- How would the operations work with more modern architectures?
- Other than the segmentation task, have the authors evaluated the methods for classification and regression problems?
- What is the theoretical guarantee of equal performance when NaN is used in the place of a regular convolution?

---

### Official Review · Reviewer_S15e · 2024-11-04

**Soundness:** 3
**Presentation:** 2
**Contribution:** 3
**Rating:** 5
**Confidence:** 4

**Summary:**

This paper presents NaN Pooling and NaN Convolution as novel methods to accelerate convolutional neural networks (CNNs) inference, specifically targeting U-Net-based models commonly used in neuroimaging, such as FastSurfer. The primary innovation lies in identifying numerically unstable voxels (often numerical noise) and replacing them with NaN values, allowing the model to skip irrelevant computations. Experimental results on the FastSurfer model demonstrate significant reductions in computational load (up to 44% of convolution operations skipped), while accuracy remains largely unaffected.

**Strengths:**

1.	Innovative use of NaN values: The approach of using NaNs to skip computations on irrelevant voxels is novel and well-aligned with the inherent characteristics of neuroimaging data, where background regions often contain redundant information.
2.	Good theoretical foundation: The paper rigorously explains the source of numerical instability in max pooling, backed by solid numerical analysis and an IEEE-standard approach to represent insignificant values as NaNs.
3.	Demonstrated efficiency gains: The empirical results convincingly demonstrate substantial computational savings, with skipped operations improving up to 69.3% in certain model layers. This is a practical advancement, particularly beneficial for large-scale neuroimaging tasks as well as other 3D medical image analyses.
4.	Detailed Experiments: The paper provides comprehensive experiments across different anatomical planes (axial, coronal, and sagittal) and a detailed analysis of NaN Pooling and Convolution's effects on FastSurfer’s efficiency and accuracy.
5.	Reproducibility: The paper contains enough implementation details that would enable reproducibility, such as NaN threshold parameters and CPU-based adjustments for PyTorch, supporting the future adoption and testing of this approach in real-world applications.

**Weaknesses:**

1.	Limited real-world impact on runtime: Although the method skips significant computations, there is no reported direct improvement in runtime, which reduces its practical appeal (as the authors rightly discuss in the conclusion). Future work should focus on addressing hardware and framework optimizations to convert computational savings into time efficiency.
2.	Data and model-specific application: The approach has been validated primarily on the FastSurfer model, which might limit generalizability. Moreover, the model has been validation only for a single dataset. NaN Pooling and Convolution may not directly transfer to models or tasks where background regions are less prevalent.
3.	Accuracy deviation in certain regions: In regions like the cerebellum, the NaN-modified FastSurfer model showed increased variability where segmentation accuracy slightly declined.
4.	Potential overhead from NaN management: The reliance on CPU-based PyTorch adaptations for NaN management is a limitation, as these are not scalable to GPU-optimized frameworks, potentially hampering applicability to larger datasets or real-time processing needs.
5.	Lack of implementation for 3D convolutions: A large fraction of medical imaging modalities produces 3D images (MRI, CT, SPECT, PET). Most recent works in 3D medical image segmentation have focussed on 3D CNNs since they allow capturing information across all three spatial dimensions, preserving the anatomical context between adjacent slices. This is also evident from many of the recent medical image segmentation challenges (organized by MICCAI), where the winning solutions utilized some version of 3D architectures such as nnUNet [Isensee, et al, Nature Methods 2020], SegResNet [Myronenko, et al, arXiv:2209:10809 (2022)], or SwinUNETR [Hatamizadeh, et al, arXiv:2201.01266v1 (2022)]. This work implements their method only for 2D CNNs which limits their broader applicability for 3D medical image segmentation.
6.	Lack of comparison to other baselines: No comparison were made to other similar methods for medical image segmentation that implement “sparsification” of data for reducing computational costs. Some of these include sparse CNN [Li, et al, 10.36227/techrxiv.19137518.v2], and dictionary learning and sparse coding [Tong, et al, NeuroImage, Vol 76 (2013)].

**Questions:**

1.	Runtime vs. computation savings: The method improves computational load by reducing convolution operations, but this does not translate directly into runtime improvements. Could the authors clarify how this approach could be adapted for GPUs or frameworks that leverage sparse matrix operations, where actual runtime gains might be realized?
2.	Implementation complexity and overheads: Given that CPU-specific adaptations were needed to manage NaNs, it would be helpful if the authors addressed whether integrating NaN Pooling and Convolution could lead to performance overheads or memory inefficiencies, particularly when deploying across high-performance computing clusters.
3.	Threshold Sensitivity Analysis: While the paper discusses threshold values of 1 and 0.5, it does not delve deeply into how threshold adjustments impact model performance and computational efficiency. Would intermediate values provide a better balance, especially in regions with high anatomical complexity?
4.	Generalizability beyond neuroimaging: The current study is highly specific to neuroimaging data with extensive background areas. How well would NaN Pooling and Convolution perform on datasets with less prominent background noise or in tasks that do not involve significant spatial redundancy?
5.	Statistical testing: In lines 348-351, the authors claim that Nan-FastSurfer performs similar to the default model in terms of Dice Loss difference, although the t-test on the difference shows significant differences (Figure 3). Does this mean that NaN-FastSurfer performs significantly worse than default FastSurfer? What significance level was chosen for this hypothesis test? Moreover, in some places in the text, the authors have mentioned that their proposed method improves computation efficiency with an equivalent performance on the DiceLoss metric. Given the fact that the authors were trying to establish an equivalence (rather than a significant difference), shouldn’t their hypothesis test be the test of equivalence such as the Two-One sided t-test (TOST) for equivalence [Lakens https://doi.org/10.1177/1948550617697177 (2017)] instead of a test of significant difference?
6.	Include a schematic: This work can also significantly benefit from the addition of a diagram/schematic showing the approach of Nan Pooling and Convolution operations. Basically, all equations on pages 2-4 can be represented as diagrams/schematics so it becomes easier to understand the details of the paper. This can be added to the main text or appendix.
7.	The paragraph in Line 162-165 seems repetitive. You can remove this paragraph.

---

### Author Response · Authors · 2024-11-27
**Rebuttal Revision**

We sincerely thank the reviewers for their thorough and insightful comments, which have significantly contributed to improving the clarity and quality of our work. We deeply appreciate the time and effort invested in providing constructive feedback. In this revised version, we have addressed the major points raised, including:
* Adding Figure 1 to address Q6 by Reviewer S15e
* Conducting additional experiments with MNIST to increase methods’ generalizability as mentioned by all reviewers
* Adding an appendix illustrating the numerical instability investigation to address weakness 1 by Reviewer 3VPG
* Addressing comments about clarity expressing concepts throughout manuscript from all reviewers

We believe these changes have strengthened the manuscript and provided a more comprehensive understanding of our contributions. Thank you again for your valuable input and for helping us enhance the impact of our research.

---

### Meta-Review · Area_Chair_gjuL · 2024-12-07

**Metareview:**

The paper proposes NaN pooling and convolution for improving the efficiency of U-Nets. The methods identifies and skips operations on irrelevant data.

All four reviewers recommend to reject the paper. Weaknesses are that the comparisons are unclear and insufficient, and the method and evaluation being narrow. I agree with the reviews and recommend to reject the paper. The reviews offer several suggestions and hints on how the paper can be improved and resubmitted in the future.

**Additional Comments On Reviewer Discussion:**

Weaknesses are that the comparisons are unclear and insufficient, and the method and evaluation being narrow

---

### Decision · Program_Chairs · 2025-01-22

Reject